# From Beach to the Bedside: Harnessing Mitochondrial Function in Human Diseases Using New Marine-Derived Strategies

**DOI:** 10.3390/ijms25020834

**Published:** 2024-01-09

**Authors:** Serena Mirra, Gemma Marfany

**Affiliations:** 1Stazione Zoologica Anton Dohrn, Department of Biology and Evolution of Marine Organisms, Villa Comunale, 80121 Naples, Italy; serena.mirra@szn.it; 2Departament of Genetics, Microbiology and Statistics, Universitat de Barcelona, Avda. Diagonal 643, 08028 Barcelona, Spain; 3Centro de Investigación Biomédica en Red Enfermedades Raras (CIBERER), Instituto de Salud Carlos III (ISCIII), Universitat de Barcelona, 08028 Barcelona, Spain; 4Institute of Biomedicine (IBUB, IBUB-IRSJD), Universitat de Barcelona, 08028 Barcelona, Spain

**Keywords:** mitochondria, disease, therapy, marine natural products, marine organisms

## Abstract

Mitochondria are double-membrane organelles within eukaryotic cells that act as cellular power houses owing to their ability to efficiently generate the ATP required to sustain normal cell function. Also, they represent a “hub” for the regulation of a plethora of processes, including cellular homeostasis, metabolism, the defense against oxidative stress, and cell death. Mitochondrial dysfunctions are associated with a wide range of human diseases with complex pathologies, including metabolic diseases, neurodegenerative disorders, and cancer. Therefore, regulating dysfunctional mitochondria represents a pivotal therapeutic opportunity in biomedicine. Marine ecosystems are biologically very diversified and harbor a broad range of organisms, providing both novel bioactive substances and molecules with meaningful biomedical and pharmacological applications. Recently, many mitochondria-targeting marine-derived molecules have been described to regulate mitochondrial biology, thus exerting therapeutic effects by inhibiting mitochondrial abnormalities, both in vitro and in vivo, through different mechanisms of action. Here, we review different strategies that are derived from marine organisms which modulate specific mitochondrial processes or mitochondrial molecular pathways and ultimately aim to find key molecules to treat a wide range of human diseases characterized by impaired mitochondrial function.

## 1. Introduction

Mitochondria are essential keepers of eukaryotic cell homeostasis, being at the crossroads of a plethora of key cellular pathways. Mitochondria play a crucial part in energy production (bioenergetics), fatty acid and amino acid metabolism regulation, intracellular calcium (Ca^2+^) and magnesium (Mg^2+^) modulation, and the triggering of cell death, among other processes. All these bio-functions are finely tuned according to genetic and environmental conditions through numerous control mechanisms that sustain optimal mitochondrial performance.

Mitochondria are double-membrane organelles that consist of an outer membrane (OMM) and an inner membrane (IMM), delimiting the inter membrane space (IMS) and the matrix. The IMM invaginates well into the matrix to form a high-membranous surface that harbors the oxidative phosphorylation system (OXPHOS) [1]. The OXPHOS is composed of electron transport chain (ETC) complexes I-IV (CI-IV), plus an ATP synthase (also called complex V, CV). ETC transfers electrons from reduced substrates (e.g., Nicotinamide Adenine Dinucleotide (NADH), Flavin Adenine Dinucleotide (FADH2), and Cytocrom C (CytC)), to molecular oxygen and performs the redox reactions involved in cellular respiration while generating the proton motive force used by CV to synthesize ATP. ETC represents a major source of ROS within the cell, as 1–2% of the electrons inevitably leak out and reduce surrounding molecular O_2_ to ROS superoxide (O_2_•^−^) [2].

The activity of mitochondrial as well as cytosolic molecular systems’ so-called antioxidants, such as superoxide dismutases, catalases, and glutathione/glutathione peroxidase, allow for the maintenance of redox homeostasis. When the rate of intracellular ROS production outweighs the antioxidant capacity of the cell, oxidative stress triggers a deleterious cascade of molecular events, eventually leading to cell death [3]. The IMM also harbors the mitochondrial permeability transition pore (mPTP), a large protein complex that, when opened, mediates the release of CytC and other apoptotic molecules from mitochondria into the cytoplasm. On the other hand, OMM acts as an interface between mitochondria and the cytosol. OMM deals with the transport of ions and metabolic molecules (such as ATP), carries all the proteins and molecules involved in mitochondrial dynamics (see below), and is involved in a variety of intracellular signaling pathways, such as the intrinsic apoptosis pathway. The antiapoptotic proteins Bcl-2 and Bcl-xL reside in the OMM, while the proapoptotic Bcl-2 proteins (Bad, Bid, Bax, and Bim) reside in the cytosol and translocate to the OMM following death signals to promote the release of CytC [4]. As an example, Bad translocates to mitochondria and forms a proapoptotic complex with Bcl-xL. Upon release from mitochondria, CytC binds Apaf-1 and forms an activation complex with caspase-9, triggering caspase cascade and apoptosis. Prior to– or simultaneously with CytC release and caspase activation, the mitochondrial network undergoes a deep remodeling and fission into fragmented units. Importantly, blocking this remodeling inhibits CytC release and delays cell death; thus, mitochondrial fission seems to be a pre-requisite step in the intrinsic apoptosis pathway, linking cell fate decisions to the morphological remodeling of mitochondria [5].

Mitochondria take shape as a complex and dynamic network constantly remodeled by proteins involved in fusion, fission, transport, quality control, and contact with other organelles, together known as mitochondrial dynamics (Figure 1). Mitochondrial fusion in mammals is catalyzed by dynamin-related proteins with GTPase activity, namely the proteins MITOFUSINS1/2 (MFN1 and MFN2) and optic atrophy gene 1 (OPA1). MFNs are located in the OMM, whereas OPA1 is located in the IMM and the IMS. In addition, MFN2 specifically mediates the contact between mitochondria and the endoplasmic reticulum (ER), ensuring the maintenance of mitochondrial metabolism, insulin signaling, and energy homeostasis. OPA1 is produced in eight distinct OPA1 isoforms in humans, which are further processed into long and short forms (L-OPA1 and S-OPA1) [6,7]. L-OPA1 is integral within the IMM, whereas S-OPA1 is located in the IMS. The levels and activity of OPA1 are regulated by two endomembrane proteases: the peptidases OMA1 and YME1L1. In mitochondria, S-OPA1 isoforms are produced via the processing at site S2 by YME1L. Other short isoforms may be obtained by the cleavage at site S1, which is regulated by OMA1 activity. While MFNs catalyze the OMM fusion, OPA1 facilitates IMM fusion by means of a GTP-coupled reaction and interaction with mitochondrial cardiolipin [8]. OMA1 activity, which is induced in dysfunctional mitochondria, results in the inhibition of mitochondrial fusion [9]. An additional mechanism of regulation of OPA1 function is provided by the action of prohibitin (PHB) in the IMM. Rings of PHB associate with m-AAA proteases and OMA1, sequestering OMA1 from its substrates [10]. In dysfunctional mitochondria, these complexes would be disassembled, allowing OMA1 to access its substrates, then induce mitochondrial fragmentation. In addition to controlling IMM fusion, OPA1 regulates cristae integrity and cristae junction maintenance, having an effect on cellular bioenergetics, mtDNA maintenance, and mitochondrial CytC release [11,12].

On the other hand, mitochondrial fission is catalyzed by Dynamin-related protein 1 (DRP1), Fission 1 homolog protein (FIS1), and mitochondrial fission factor (MFF). Mitochondrial fission requires the recruitment of the cytosolic GTPase DRP1 to the outer mitochondrial membrane, where it wraps around and constricts the mitochondrial tubule to mediate fission. In this context, FIS1 and MFF, located on the outer mitochondrial membrane, act as receptors that recruit DRP1 to the mitochondrial surface [13]. 

The molecular pathways responsible for regulating mitochondrial fusion/fission are strictly linked by a cellular quality control mechanism and are considered quality control mechanisms that preserve mitochondrial health. Indeed, when the mitochondrial membrane potential (Δψm) is slightly altered, mitochondria fuse with others, allowing each mitochondrion to compensate for its own defects by mixing molecular components and dilute damaged molecules. On the other hand, fusion capacity is inhibited in mitochondria when damage is severe, which facilitates their elimination by mitophagy, the mitochondrial-specific autophagy pathway (discussed below) [14,15]. 

Mitochondrial fusion inhibition is achieved by either inactivating fusion or activating fission. As an example, the proteolytic-mediated processing and inactivation of OPA1 has been described to inhibit IMM fusion, along with the subsequent segregation of damaged organelles. It is likely that segregated and small mitochondria are easier to be degraded compared to larger elongated mitochondria. One of the most well-studied mechanisms of mitophagy involves *PINK1* and *PARKIN*, two genes associated with rare genetic forms of Parkinson’s disease [16]. PINK1 is a serine–threonine kinase normally undetectable in most cells because it is cleaved by Presenilin-associated rhomboid-like protein (PARL) and targeted for degradation by mitochondrial peptidases, thus preventing its accumulation on healthy mitochondria [17]. In depolarized mitochondria, PINK1 is no longer cleaved and becomes stabilized on the outer mitochondrial membrane, where it phosphorylates PARKIN, inducing its recruitment to mitochondria. Mitochondrial PARKIN is an E3 Ubiquitin ligase that ubiquitinates several OMM proteins. For example, the ubiquitination and proteasomal degradation of MFNs results in mitochondrial fragmentation. Moreover, the ubiquitination of autophagy receptor proteins promotes the engulfment of mitochondria by autophagosomes, which then merge with lysosomes for degradation [18]. Functional deficits in one of the multiple pathways regulating mitochondrial biology are likely to have pathological molecular outcomes, leading to detrimental results with respect to cell viability and triggering the onset of many human diseases. Consequently, the key effector proteins mediating mitochondrial integrity, bioenergetics, and dynamics have emerged as a novel class of drug targets, defining a very promising field in pharmacology and biomedicine [19,20].

The World Register of Marine Species (WORMS) lists 243.417 accepted species living in very diverse ecosystems within the marine environment which are frequently characterized by conditions of light, oxygenation, pressure, and/or temperature that are challenging or prohibitive for most animal species. Strong selective pressures during evolution promoted the generation of different species of marine organisms adapted to a wide range of environmental conditions [21]. This scenario provides an abundance of physiological, biochemical, and molecular innovations which both biomedicine and biotechnology can draw from to improve the quality of human life [22,23]. Biomedicine is frequently inspired by the extraordinary abilities of marine organisms, such as their capacity to regenerate organs or even, in many cases, the whole organism (e.g., sea cucumbers, sea urchins, starfish, planaria, or cephalopod mollusks). Moreover, marine (blue) biotechnology relies on biomolecules produced by marine organisms (mainly invertebrates) such as mucus, adhesive gels, polysaccharides, venoms, and light-producing molecules. 

Several marine-derived drugs have been highlighted to target mitochondria or mitochondrial signaling pathways and could thus counteract pathological processes in many human diseases, including neurodegenerative diseases, metabolic disorders, cancer proliferation, and metastasis.

Here, we review the recent advances in biomedical strategies to treat human diseases by harnessing mitochondrial function and/or dynamics using marine resources. Therefore, we will exclusively focus on biomolecules derived from marine organism therapeutic strategies that target mitochondria, ranging from mitochondrial biogenesis to mitochondrial degradation (summarized in Table 1 and Table 2 and Figure 2).

## 2. Modulators of Mitochondrial Biology Derived from Marine Resources

### 2.1. Enhancing Mitochondrial Biogenesis

Mitochondrial biogenesis is the process by which cells increase both the number of mitochondria and their mitochondrial mass. During mitochondrial life cycle, fission occurs in damaged mitochondria to facilitate mitochondrial removal by mitophagy. However, fission also represents a strategy of mitochondrial self-duplication in which new mitochondria are formed by preexisting ones. In fact, the current view is that mitochondria are never formed de novo. Thus, in mammalian cells, mitochondrial biogenesis must be finely regulated by both nuclear and mitochondrial genomes [71]. This process is triggered by physiologic stimuli including physical exercise, calorie restriction, temperature, and muscle myogenesis. 

Peroxisome proliferator-activated receptor (PPAR) γ—coactivator-1 (PGC1-α) has been identified as a master regulator for mitochondrial transcription and translation mechanisms. PGC-1α co-activates nuclear respiratory factors 1 and 2 (NRF1 and NRF2) and, subsequently, mitochondrial transcription factor A (TFAM). The activation of the PGC-1α–NRF–TFAM pathway leads to the synthesis of mtDNA and proteins, and, thus, the generation of new mitochondria [71]. Moreover, SIRTUIN1 (SIRT1) deacetylates PGC1α, leading to both increased and activated PGC1α levels. AMP-activated protein kinase (AMPK) is able to modulate the levels of NAD+, causing SIRT1 activation, thus increasing mitochondrial biogenesis [72].

Many preclinical studies have reported natural compounds, from both marine and terrestrial sources, that are able to enhance mitochondrial biogenesis [73]). In the marine world, algae are an extremely vast group with a substantial reservoir of biomolecules such as bioactive peptides, long-chain polyunsaturated fatty acids (LC-PUFAs), steroids, carotenoids, polysaccharides, vitamins, and so on [74]. See Table 1 and Figure 2 for a brief summary.

LC-PUFAs are largely present in fish and have been supplied to humans by fish oil and seafood because of their beneficial effects on human health. However, the continuous increase in the global population has led to an insufficient production of LC-PUFA-enriched-products. Various marine microalgae and macroalgae are able to produce high amounts of PUFAs. The n-3 long-chain PUFA eicosapentaenoic acid (EPA, 20:5 n-3) predominates among different microalgae species from the Eustigamatophyceae and Prasinophyceae genus, while docosahexaenoic acid (DHA, 22:6 n-3) is commonly found in heterotrophic microalgal species such as *Schizochytrium* sp. and *Crypthecodinium cohnii*. Thus, the possibility of using microalgae to produce n-3 PUFA-rich oils has been explored, and some species have already been used commercially to produce n-3 PUFA, in particular EPA- and/or DHA-rich oils. EPA and DHA have been shown to induce metabolic changes in several tissues in both mice and humans [75]. The diet-mediated administration of marine-derived EPA and DHA upregulates PGC1-α and NRF1 in C57BL/6J epididymal fat and, consequently, increases mitochondrial biogenesis [75]. Moreover, EPA/DHA showed an antiadipogenic effect in [24] by decreasing lipogenesis and obesity. 

Algae-derived marine polysaccharides have been shown to be essential in numerous applications in the pharmaceutical, nutraceutical, and cosmetics fields due to their promising bioactive properties [76]. Brown algae contain a wide variety of acidic polysaccharides (e.g., alginates, fucoidans, and ulvans), whereas little research has been conducted on microalgal polysaccharides. Alginic acid is a linear polymer formed by chains of β(1→4)-linked d-mannuronic acid and α(1→4)-linked l-guluronic acid, and it is mainly extracted by *Laminaria* algae; it has been used in various human health applications, such as protein delivery and cell encapsulation. Moreover, several biological activities have been described, including neuroprotective, antitumoral, and antidiabetic roles [77]. Two different studies from the same group showed the positive effects of oligomannuronate (OM) and two kinds of oligomannuronate-chromium (III) complexes (OM2 and OM4) from the marine brown alga *Laminaria japonica* on the pathways regulating mitochondrial biogenesis and metabolism. They showed that both OM2 and OM4 enhance glucose uptake in skeletal muscle C2C12 cells without obvious toxicity. This effect is greater in OM2 than OM4. Both of them were distributed to the mitochondria and increased the expression of PGC-1α, glucose transporter 4 (GLUT4), and the insulin receptor (IR) by activating both the PI3K/Akt and AMPK pathways [25]. Molecular mechanisms of action were further investigated in differentiated 3T3-L1 adipocytes, where OM2 was found to stimulate the AMPK-PGC1α pathway, induce mitochondrial biogenesis, improve mitochondrial function, and reduce lipid accumulation as a consequence of enhanced fatty acid β-oxidation and increased adipose triglyceride lipase (ATGL) protein expression [25]. The authors concluded that OM2 might represent a key molecule for targeting type 2 diabetes alterations by enhancing mitochondrial biogenesis and function. 

Molecules able to regulate mitochondrial biogenesis have been also isolated in the oceanic invertebrate sea cucumber, an organism under the phylum Echinodermata and the class Holothuroidea. Sea cucumber is utilized as a traditional medicine in several Eastern countries because of the beneficial proprieties of its peptides (SCPs), particularly their antifatigue capacity. SCP treatment has been reported to improve ATP turnover, mitochondrial biogenesis, and overall mitochondrial quality in mice [78]. Recently, two SCPs with low hydrolysis (SCP-1) and high hydrolysis (SCP-2) derived from *Acaudina leucoprocta*, were investigated. Both SCP-1 and SCP-2 significantly improved exercise performance and exerted antifatigue effects in mice in comparison to the negative control group. The antifatigue effects provided by the SCP-2 treatment were greater. The molecular mechanisms underlying these effects were identified, along with their impact on the regulation of the NRF2 and AMPK/PGC1-α signaling pathways. At the cellular level, the main phenotypic outcomes arising from these treatments included the significant inhibition of oxidative stress and the improvement of mitochondrial biogenesis and function. The authors concluded that SCPs might be a novel functional nutritional additive to treat fatigue throughout the modulation of mitochondrial biogenesis [26].

### 2.2. Modulating Mitochondrial Dynamics 

The modulation of mitochondrial shape/dynamics represents a strategy that has been successfully exploited over the years to reduce neurodegeneration [79] or ischemia-induced damage [80]. Several marine organism-derived molecules represent promising candidates to act likewise (summarized in Table 1 and Figure 2), extending their range of action to cancer. 

Aurilides are certainly the best known marine compounds for the regulation of mitochondrial dynamics. Aurilide A was isolated in 1996 from the Japanese sea hare *Dolabella auricularia* and was described as a potent cytotoxic cyclodepsipeptide [28,81]. From its identification, several related compounds with very similar features have been identified, including but not limited to Aurilide B and C from the cyanobacterium *Lyngbya majuscule* [29]; kulokekahilide-2 from the cephalaspidean mollusc *Philinopsis speciosa* [30]; and lagunamides A, B, C, and D’ from different Papua New Guinea marine cyanobacteria *Lyngbya majuscules* [29,31,32,82,83,84].

In 2011, Sato et al. described the molecular mechanism by which aurilide exercises its cytotoxic activity, linking this molecule to the pathway of mitochondrial dynamics and cristae remodeling (controlled by PHB and OPA1) [27,85]. As stated before, PHB interacts with the m-AAA protease complex, which is involved in OPA1 processing [86]. Aurilide accumulates in mitochondria and induces the disassembly of PHB from m-AAA, thus accelerating OPA1 processing (the conversion of L-OPA1 to S-OPA1) and mitochondrial fragmentation. Importantly, OPA1 also regulates cristae integrity and cell death: a high-molecular-weight oligomer composed of both S-OPA1 and L-OPA1 retains CytC in mitochondria, thus preventing apoptosis. By inhibiting PHB, aurilide induces the processing of OPA1, the destabilization of its oligomers, and enhance the release of CytC [27]. Preclinical studies have already described aurilides to be active against several cancer cell lines, such as a cervical cancer cell line (HeLa), the Neuro-2a mouse neuroblastoma cell line, and NCI-60, a cell panel created by the U.S. National Cancer Institute to characterize drug sensitivity markers [29]. Moreover, lagunamides from the cyanobacterium *Lyngbya majuscule* exhibit cytotoxicity via mitochondria-mediated apoptosis toward colorectal carcinoma cells (HCT-8), lymphoma cells (P388), lung adenocarcinoma cells (A549), ovarian cancer cells (SK-OV-3), and prostatic adenocarcinoma cells (PC-3), even if a direct functional correlation with master regulators of mitochondrial dynamics has yet to be provided [87]. The latter is common for several marine-derived compounds that have been shown to ameliorate the organization of the mitochondrial network in preclinical studies, even though the precise mechanism of action on mitochondrial dynamics remains unknown.

Piscidin-1, a 22-residue cationic peptide isolated from the mast cells of a hybrid striped bass, was initially described as a potent antimicrobial compound that is also able to inhibit the migration of fibrosarcoma cells (HT1080) [33,34]. Piscidin-1 has recently been described to exert growth inhibition on various cancer cells, generating interest among those involved in the development of new chemotherapeutic drugs. Piscidin-1 treatment has been shown to induce a reduction in the expression levels of MFN1, MFN2, and OPA1 and an increase in the expression levels of DRP1 and FIS1 proteins in an osteosarcoma cell line (MG63). Other mitochondrial phenotypes induced by piscidin-1 include those related to mtROS production, OXPHOS protein downregulation, mitochondrial dysfunction, and apoptosis [34]. Again, the detailed molecular mechanism of action of this peptide has not been described yet.

Xyloketal B, produced by the mangrove fungus *Xylaria* sp. (no. 2508) from the South China Sea, is a metabolite that has bene extensively studied in both in vitro and in vivo disease models. Xyloketal B and its derivatives exhibit cytoprotective effects in cardiovascular, neurodegenerative, and non-alcoholic fatty liver disease models, mainly acting as an antioxidant [35]. The mitochondrial mechanism underlying the protective potential of Xyloketal B has been described in an in vitro model of ischemic stroke in PC12 cells: Xyloketal B treatment alleviated damage induced by oxygen deprivation insult via significantly increasing DRP1 expression levels so that mitochondria fragmentation was reverted and mitochondria superoxide production was reduced [88].

### 2.3. Inducing Mitophagy

Marine organisms such as algae, sea cucumber, and sponges provide a number of molecules able to regulate general autophagy that have been tested in preclinical studies as anticancer agents by either inducing autophagic cell death or preventing prosurvival autophagy processes in cancer cells, thus helping tumor cells to survive in unfavorable metabolic conditions [89]. However, only a few marine compounds are known to specifically target mitophagy, and the effects of marine drugs on mitophagy can be ascribed to alterations in general autophagy. One exception is a polyphenol recently isolated from the red alga *Polysiphonia japonica*, 5-Bromoprotocatechualdehyde (BPCA). BPCA has been described as a potential compound for the protection of β-cells, which are lost during diabetes pathogenesis. In a cellular model of palmitate (PA)-induced lipotoxicity, BPCA treatment induced several beneficial effects at the cellular level, including the preservation of PARKIN expression and the stabilization of mitochondrial morphology [36].

Notably, the interest in polysaccharides from natural sources, including marine organisms, has increased over the years due to their promising applications in the field of drug delivery. Fucoidan (FU) is a natural polysaccharide extracted from the extracellular matrix of brown algae. It can be beneficial for human health by acting as an antitumoral, antioxidation, anticoagulation, and anti-inflammatory molecule. Moreover, FU improves the water solubility of hydrophobic drugs and has been shown to be ideal for drug delivery in functional foods due to its biocompatibility and biodegradability [90]. Importantly, a wide range of nanostructures has been developed with this polysaccharide per se and in combination with other natural bioactive compounds [91]. As an example, FU nanoparticles have been prepared in combination with proanthocyanidins (PC), natural polyphenolic compounds abundant in the plant kingdom. Numerous in vitro and in vivo studies have demonstrated that PC produces many effects that could potentially be beneficial to human health. However, they showed very low bioavailability, with 90% remaining unabsorbed from the intestines. Combining FU/PCs in the form of nanoparticles (FU/PCs NPs) improved the antioxidant activity of PCs both in vivo and in vitro by acting through a mechanism involving mitophagy. FU/PCs NPs activated mitophagy by increasing the expression of both PINK1 and PARKIN and inhibited the release of mtDNA caused by cisplatin induced-damage [37].

### 2.4. The Regulation of Cell Death

Apoptosis is one of the better known forms of regulated cell death, which can be triggered by the activation of cell-surface death receptors like FAS (the extrinsic pathway) or the activation of proapoptotic proteins belonging to the BCL-2 family, which cause the permeabilization of the MOM (the intrinsic pathway). Both pathways converge on the activation of caspases, thus triggering cell demise in the so-called execution phase of apoptosis. 

Marine organism-derived compounds are promising modulators of cell death because they target molecular mechanisms that regulate apoptosis. Accordingly, most of these marine-derived compounds have been studied for their ability to suppress cancer growth and progression. Some of these compounds are already commercially available, such as Cytarabine (Ara-C, Cytosar), trabectedin (ET-743, Yondelis), eribulin mesylate (E7389, Halaven), and brentuximab vedotin (SGN-35, Acentris), with more anticancer drugs in phase I, II, or III clinical trials [92]. However, none of these compounds focus their actions specifically on mitochondria-related pathways.

Due to the pivotal role of mitochondria in regulating tumorigenesis and cancer progression, the interest in marine-derived anticancer agents has grown during the last decade concomitantly with the development of novel biotechnological approaches to culture marine organisms and extract bioactive compounds with meaningful biomedical interest. Harnessing marine-derived agents to target mitochondrial function will be the next and inevitable challenge, which, when conquered, will provide key tools for the management of uncontrolled cell division, a sign of tumorigenesis. 

Marine flora include bacteria, actinobacteria, cyanobacteria, fungi (altogether known as microflora), microalgae, seaweeds, mangroves, and other, constituting over 90% of the oceanic biomass. Marine microflora produce bioactive toxins and metabolites, and micro and macroalgae are widely used as nutraceuticals and food supplements because they are sources of vitamins, polysaccharides, omega-3, carotenoids, and other molecules [93]. Many bioactive molecules produced by these organisms have largely been reported to have anticancer activity due to their antioxidant, anti-inflammatory, and immunomodulatory proprieties. Frequently, the mechanisms of action of these compounds are associated with mitochondrial pathways, as unveiled by preclinical studies performed in cell cultures or in animal cancer models. Most promising marine bioactive compounds derived from marine flora assayed in cancer models and/or tested in patients as anticancer agents have been outstandingly reviewed elsewhere [88,94].

Importantly, 11 out of the 15 marine-derived drugs approved for cancer treatment came from marine fauna, particularly invertebrate animals such as sponges, tunicates, and mollusks. They have the following main targets: DNA polymerases, triglyceride-synthesizing enzymes or microtubules, among others (https://www.marinepharmacology.org/approved (accessed on 4 January 2024)) [95]. Here, we will focus on marina fauna that produce compounds associated with a direct action on mitochondria (summarized in Table 2).

#### 2.4.1. Marine Sponges

Several marine compounds containing quinone are known to have antiproliferative effects by targeting mitochondria. Ilimaquinone, a sesquiterpene quinone isolated from *Halichondria* marine sponges, increases caspase activity, ROS production, and Δψm loss in MCF-7 and MDA-MB-231 human breast cancer cell lines [38]. The polycyclic quinone-type metabolite xestoquinone, extracted from *Petrosia* sp., decreased the growth of leukemia cancer cells via the ROS-induced suppression of heat shock protein-90 in [39]. Moreover, Renieranycin M (RM, a bistetrahydro-isoquinolinequinone isolated from *Xestospongia* sp.) has been reported to have antiproliferative effects in human lung cancer through a mitochondria-dependent pathway [40,41].

The marine indole imidazole alkaloids Nortopsentins A-C, from the sponge *Spongosorites ruetzleri*, have a cytotoxic effect on leukemia P388 cells [43]. A similar effect is provided by two analogues of nortopsentin, 5-bromo-1-methyl-3-[2-(1H-pyrrolo[2,3-b]pyridin-3-yl)-1,3-thiazol-4-yl]-1H-pyrrolo[2,3-b]pyridine (T1) and 3-[2-(5-Bromo-1H-indol-3-yl)-1,3-thiazol-4-yl]-7-chloro-1H-pyrrolo[2,3-c]pyridine (T2), which have cytotoxic effects on several human cancer cell lines. Moreover, T1 activates the mitochondria-mediated apoptotic pathway, inducing cell cycle arrest at the G2/M phase and inhibiting CDK-1 activity in HCT 116 colon cancer cells, whereas T2 induces the accumulation of autophagic vacuoles at low concentrations while triggering apoptosis only at high concentrations [42].

Other sponge-derived alkaloids that induce apoptosis by targeting mitochondria include Manzamine A (from *Haliclona* sp., *Xestospongia* sp., and *Pellina* sp.) and aplysinopsins (from several sponges, such as *Thorecta* sp., as well as corals). Both have been shown to decrease the level of Bcl-2 and cause Δψm loss. Manzamine A enhanced the activity of caspase-3 and caspase-7 and induced the release of CytC in HCT116 cells in [44], while aplysinopsins and the aplysinopsin analog EE-84 reduced the intake of oxygen and cell proliferation in K562 cells in [45].

Irciniastatin A is a pederin-type compound from the marine sponge *Ircinia ramosa* which induces the production of ROS in mitochondria, thus activating the stress-induced protein kinases JNK and p38 and leading to apoptosis in human leukemia Jurkat cells [46].

A furanoterpenoid isolated from irciformonin B found in a marine sponge, 10-acetylirciformonin B, blocked Bcl-2 and Bcl-xL, enhanced Bax expression, stimulated the release of CytC, and increased the levels of ROS in mitochondria in [96]. Additionally, 10-acetylirciformonin B has been shown to exhibit antitumor properties in leukemia HL60 cells via promoting DNA destruction and apoptosis through several signaling pathways; notably, 10-acetylirciformonin B induced caspase-3 and caspase-9 activity and decreased topoisomerase II protein production [96].

An extract from the marine sponge *Petrosaspongia mycofijiensis* yielded mycothiazole, which inhibits hypoxic HIF-1 signaling and causes mitochondrial dysfunction in human breast tumor T47D cells. Mechanistic studies have revealed that mycothiazole selectively suppresses mitochondrial respiration at Complex I [47].

A dipeptide Cyclo(-Pro-Tyr) (DP) isolated from the marine sponge *Callyspongia fistularis* stimulated the production of ROS, increased the rate of Bax/Bcl-2 and caspase-3, and facilitated the release of CytC in HepG2 cell lines [48]. A similar effect was shown for the rare deep-sea marine sponge *Monanchora pulchramarine*-derived Urupocidin A, a bicyclic guanidine alkaloid that targets mitochondria inducing mitochondrial membrane permeabilization, the release of cytotoxic mitochondrial proteins to the cytoplasm, ROS production, the activation of caspase-9 and caspase-3, and subsequent apoptosis in prostate cancer cells [49].

The marine compounds 3,5-dibromo-2-(20,40-dibromophenoxy)-phenol and 3,4,5-tribromo-2-(20,40-dibromophenoxy)-phenol, extracted from the marine sponge *Dysidea* sp., increased the levels of phosphorylated AKT in [50], having anticancer effects in pancreatic cancer cells. The 3,5-dibromo-2-(20,40-dibromophenoxy)-phenol impaired mitochondrial function by targeting Complex II of the mitochondrial electron transport chain [50]. The sponge *Dysidea avara*, which produces the sesquiterpene lactone santonin, also decreased Δψm, enhanced ROS production, activated caspase-3, and induced CytC relase in all B-lymphocytes [51].

Another cytotoxic compound derived from sponges that act via mitochondrial dynamics is the toxin papuamine (from *Haliclona* sp.), which has shown activity on human MCF-7 breast cancer cells [52].

#### 2.4.2. Corals

2-ethoxycarbonyl-2-β-hydroxy-A-nor-cholest-5-ene-4one (ECHC), isolated from butanol extracts of the hexacoral *Acropora formosa*, enhances mitochondrial-mediated apoptosis in non-small-cell lung cancer cells (A549). ECHC treatment increases the expression of antiapoptotic proteins (such as p53, tumor necrosis factor-α (TNF-α), interleukin-8 (IL-8), Bcl-2, MMP-2, and MMP-9) and enhances ROS generation and CytC release from mitochondria in human non-small-cell lung cancer cells [53].

Methyl 5-[(1E,5E)-2,6-Dimethyl octa-1,5,7-trienyl] furan-3-carboxylate (MDTFC), a furano-sesquiterpene extracted from the soft coral *Sinularia kavarittiensis*, has been reported to induce apoptosis via increasing the rate of Bax/Bcl-2 production and CytC release, as well as decreasing Δψm in THP-1 (human monocytoid) cells [54].

Other cytotoxic compounds derived from the coral *Sinularia flexibilis* include Sinularin, which induces apoptosis through mitochondria dysfunction and inactivating the pI3K/Akt/mTOR pathway in gastric carcinoma cells, namely AGS and NCI-N87, and 11-dehydrosinulariolide, which acts through mitochondrial-related pathways to induce apoptosis in both hepatocellular carcinoma SK-HEP-1 and melanoma cells [55,56]. Interestingly, 11-dehydrosinulariolide has also been described to upregulate the Akt/PI3K pathway and protect cells against 6-hydroxydopamine (6-OHDA)-mediated damage in neuronal cells via increasing cytosolic or mitochondrial DJ-1 expression. The latter activates the downstream Akt/PI3K, p-CREB, and Nrf2/HO-1 pathways, with a neuroprotective role both in vitro and in vivo in zebrafish and rat models of Parkinson’s disease [97].

#### 2.4.3. Marine Tunicates

Aplidin is a cyclic depsipeptide derived from the marine tunicate *Aplidium albicans* that is able to induce apoptosis and activate p38 and MAPK signaling pathways. This compound is now produced synthetically and has been shown to act on mitochondria by producing ROS and decreasing Δψm and ATP concentrations, both in vitro and in vivo. Its citotoxic effect has been demonstrated in leukemia lymphoma models, as well as in many hematological and solid cancers [57,58].

Compounds generated by the marine tunicate of the ascidian *Phallusia nigra* showed significant antitumor activities on skin mitochondria isolated from a albino/Wistar rat model of melanoma. N-hexane, diethyl ether, and methanolic extracts from *P. nigra* enhanced ROS production and mitochondrial swelling, reduced Δψm, and induced the release of CytC from mitochondria [59].

Mandelalides are marine polyketides that are cytotoxic in their glycosylated forms. Mandelalides A and B, isolated from a new ascidian species of the genus *Lissoclinum*, showed potent cytotoxicity to human NCI-H460 lung cancer cells and mouse Neuro-2A neuroblastoma cells [60]. The glycosylated forms of the newly characterized mandelalide E reisolated natural mandelalides B and C and synthetic mandelalide A and reduced the viability of human NCI-H460 lung cancer, HeLa, U87-MG glioblastoma, and HCT116 colon cells by inhibiting aerobic respiration [61]. Moreover, mandelalide A inhibits ATP synthase activity from isolated mitochondria and triggers caspase-dependent apoptosis in HeLa cells [98].

CS5931, extracted from sea squirt *Ciona savignyi*, has shown potent antiproliferative and proapoptotic activities. This compound has been shown to act on caspase-3, caspase-9, CytC, Bax, and Δψm in HCT-8 colon cancer cells [62] (a summary is provided in Table 2 and Figure 2).

#### 2.4.4. Holothuroids

As stated before, sea cucumber peptides (SCPs) are known to provide numerous health benefits. Additionally, compounds isolated from holothuroids are cytotoxic against several types of cancer. Several mechanisms of toxicity against cancer by sea cucumber-derived compounds have been described, including mitochondrial targeting [99]. Echinoside A and ds-echinoside A, which are glycosylated triterpenes isolated from *Pearsonothuria graeffei*, induced apoptosis via the intrinsic mitochondrial pathway in HepG2 cells, and providing treatments involving these two compounds to mice bearing H22 hepatocarcinoma tumors reduced tumor weight by 49.8% and 55.0%, respectively.

Stichoposide C, isolated from *Thelenota anax*, has been shown to induce apoptosis in a dose-dependent manner in human leukemia and mouse colorectal cancer cells. Its mechanism of action includes the activation of Fas, caspase-3, and caspase-8; the cleavage of Bid; and mitochondrial damage [64].

Methanolic extracts obtained from sea cucumber *Holothuria parva* have been shown to increase ROS, decrease Δψm, enhance CytC release, and induce caspase-3 activity in mitochondria isolated from an animal model of hepatocellular carcinoma [65]. Analogous results were obtained in the same study by using compounds derived from *Haliclona oculata*. This study was conducted as part of a national project that hopes to identify novel potential anticancer candidates in the Persian Gulf [65].

#### 2.4.5. Marine Mollusks

Lamellarins are a class of bioactive marine pyrrole alkaloids derived from various species and acting on multiple targets to induce cell death in several cancer cell lines. Lamellarin D, isolated from the marine mollusc *Lamellaria*, is the most studied component. The molecular mechanism by which Lamellarin D induces apoptosis has been described in p388 leukemia cells, where it increased the rate of Bax, stimulated the activity of caspase-3 and caspase-9 and decreased that of Bcl-2, and finally promoted the depolarization of mitochondria, as well as nuclear apoptosis [66,67].

The Persian Gulf marine mollusc *Turbo coronatus* has been shown to have antitumoral activity in epithelial ovarian cancer EOC cells. Cells and isolated mitochondria exposed to extract fractions of *T. coronatus* suffered an increase in ROS production, mitochondrial membrane depolarization, mitochondrial swelling, and CytC release, finally resulting in apoptosis and necrosis [68].

Similarly, conotoxins derived from *Conus textile*, a marine cone snail in Iran’s southern seas, exhibited antitumor properties by inducing the activity of caspase-3 and caspase-9, increasing Bax/Bcl-2 levels, and enhancing the production of CytC and ROS in U87MG human glioma cells [69].

A polysaccharide gleaned from the marine clam *Donax variabilis*, fraction 2.1, has been described to induce mitochondrial dysfunction in A549 cells by also inducing the activity of caspase-3 and caspase-9, increasing Bax/Bcl-2 levels, and enhancing the production of CytC, thus triggering apoptosis [70].

A summary of the marine-derived compounds described in this review, along with their reported mechanisms of action on mitochondria, is provided in Figure 2.

## 3. Gene Therapy Approaches Inspired by Marine Organism Strategies

Primary mitochondrial diseases (PMDs) form a heterogeneous group of inborn errors of metabolism caused by defects related to the mitochondrial OXPHOS system and may arise from pathogenic mutations in nuclear DNA and/or mtDNA. Conversely, secondary mitochondrial disorders (SMDs) do not arise from mutations directly impacting on OXPHOS production or function. SMDs accompany many hereditary non-mitochondrial diseases and might also be caused by non-genetic causes such as environmental stressors. Both PMDs and SMDs are caused by the mutation of a rapidly increasing group of genes encoding either OXPHOS proteins, assembly factors, proteins responsible for controlling mitochondrial dynamics and quality control, proteins involved in mtDNA replication, mitochondrial transcription and translation, cristae shaping, the biosynthesis of redox cofactors, or detoxifying pathways, among others. Currently, there is no cure for OXPHOS defects. The pathophysiological effects of OXPHOS deficiency result in two main molecular outcomes: decreased ATP production and increased ROS production and subsequent toxicity. This extremely high heterogeneity of genetic and non-genetic mitochondrial diseases poses a significant limitation to the development of specific therapeutic strategies.

Recently, marine organisms inspired a solution to bypass defects related to CIII-CIV OXPHOS components with any genetic or environmental basis. Under physiological conditions, quinols that transport electrons in the mitochondrial inner membrane are efficiently oxidized by CIII, and then electrons are transferred to oxygen via CIV. When complexes CIII and/or CIV are dysfunctional, the electron flow is interrupted, fueling the generation of ROS and misbalancing redox and metabolic homeostasis. In many lower organisms and plants, alternative components of the respiratory chain, namely alternative oxidases (AOXs), oxidize reduced CoQ (ubiquinol) by transferring electrons directly from ubiquinol to oxygen (with water generation), thereby bypassing the activity of OXPHOS CIII-CIV. AOX-mediated electron transfer dissipates free energy as heat without generating proton-motive force. Moreover, the enzyme is unaffected by cyanide, in contrast to the cyanide-sensitive CIV. The loss of AOXs in vertebrates is poorly understood, although it has been proposed that the concomitant presence of well-developed ROS signaling pathways in these organisms would prevent the activity of AOXs to be beneficial in order to maintain the overall cellular homeostasis [100]. Importantly, AOXs are activated only when the Q-pool reduction level reaches 35–40%, meaning that these enzymes provide a very attractive opportunity to bypass genetic and non-genetic CIII- and CIV-related defects by using the xenotopic expression of a molecular switch perfectly set up by evolution.

The most used AOX for therapeutic purposes is derived from the marine tunicate (sea squirt) *Ciona intestinalis* (CiAOX), which is the closest known relative to vertebrates [101]. CiAOX was initially expressed in cultured human cells, and it localized in mitochondria conferring cyanide resistance. It remains essentially inactive in healthy cells, and its catalytic activity is triggered by the highly reduced redox status of the respiratory chain and is enhanced by pyruvate accumulation [102]. Moreover, CiAOX was efficiently expressed in *Drosophila melanogaster* [103], where it rescued some OXPHOS defects [103,104,105,106,107].

These studies paved the way to perform the first studies in mouse models of several diseases, such as those caused by genetic alterations of OXPHOS function, for instance, cardiac disfunctions or lung diseases (reviewed in [108]). In 2017, a mouse knock-in of CiAOX targeting the Rosa26 locus for ubiquitous expression was generated (AOXRosa26); it essentially showed a normal physiology, but it was cyanide-resistant, and CiAOX expression conferred robust resistance to the inhibitors of the respiratory chain in the organelles [109]. The AOXRosa26 strain was subsequently crossed with several mitochondrial disease mouse models to assess the possibility of phenotype rescue and/or the prevention of the most severe disease outcomes [107,109,110,111,112,113,114]. Results from different studies highlight how CiAOX may produce very different effects depending on the mitochondrial pathology. As an example, CiAOX expression in a severely myopathic skeletal muscle-specific COX15 knockout mouse exacerbated myopathy [110]. This is likely due to the ROS signaling pathway experiencing interference, which triggers a compensatory mechanism of mitochondrial biogenesis and drives the physiological response of muscle regeneration. Importantly, the antioxidant N-acetylcysteine had a similar effect, decreasing the lifespan of KO mice. These results have large impacts on the design of therapeutic strategies, highlighting the benefits of ROS stress response and the potential hazards of antioxidant treatment.

On the other hand, CiAOX overexpression has led to beneficial results in other pathological contexts. This was the case in a study involving a mouse model of GRACILE (growth retardation, aminoaciduria, cholestasis, iron overload, lactacidosis, and early death) syndrome—causing a severe CIII deficiency—probably caused by a mutation that results in an S78G amino acid change in the BCS1L gene [115,116]. BCS1L, a mitochondrial inner-membrane protein, is a chaperone necessary for the assembly of mitochondrial respiratory chain complex III. Homozygous Bcs1lc.A232G (Bcs1lp.S78G) mice bearing the GRACILE syndrome-analogous mutation have been shown to recapitulate many of the clinical manifestations in patients, such as short survival, growth failure, progressive hepatopathy, and kidney tubulopathy [117,118,119]. When the AOXRosa26 strain [109] was crossed with the Bcs1lc.A232G mice, the transgenic progeny presented an extended median survival due to the permanent prevention of lethal cardiomyopathy [118]. Renal tubular atrophy and cerebral astrogliosis were also prevented, in contrast to liver disease, growth restriction, and lipodystrophy, thus suggesting distinct tissue-specific pathogenic mechanisms. At the cellular level, mitochondria structure and function, as well as transcriptomics and metabolomics alterations, were rescued by AOX overexpression.

Altogether, the data collected in vivo differentially support an AOX-based therapy. They have highlighted several limitations presented by CiAOX, which, in nature, exists in a cellular context that is very different from the mammalian one. In mammals, processes of tissue plasticity and remodeling rely on novel molecular mechanisms of resilience against oxidative stress, such as the ROS signaling pathway. However, it is worth noting that AOXs may facilitate very different effects depending on the tissue. Future perspectives on the topic should consider the generation of synthetically engineered AOX proteins based on those described in tunicates and tailored to specific applications. For instance, the CiAOX sequence might be improved by modifying its oxidation activity via changes either in the protein structure and/or folding, or increasing/decreasing sensitivity to specific regulators and/or modulators [120].

## 4. Conclusions, Challenges, and Future Perspectives

Marine organisms have been used since ancient times as a primary source of therapeutic biomolecules in traditional medicines. As an example, classical texts from the Ancient Greek and Early Byzantine periods describe the therapeutical properties of marine invertebrates, reflecting their common use in medical practice [121]. The cellular and molecular bases of these therapeutical actions were not known at that time. Nowadays, we know mitochondria are organelles that act as metabolic hubs of cellular signaling production and transmission [122]. They play a pivotal role in the adaptative responses to complex environmental challenges. Marine organisms are an important source of promising multi-target agents that are able to modulate mitochondrial biology and signaling pathways. Here, we reviewed molecules obtained from—and strategies inspired by—marine organisms that may be potentially used to mitigate the symptoms of all the disorders with a mitochondrial pathological basis or involving a mitochondrial-related target protein. Importantly, the marine environment still has to offer a huge and unexplored biodiversity, like that present in deep-sea environments or other very extreme/secluded places, which are species-rich habitats that have been less explored compared to more accessible sites. For instance, due to their adaptation to this extreme environment, deep-sea species have the potential to produce a totally novel group of molecules with potent biological activities, bestowing an unexplored trove of novel therapeutically strategies to alleviate or treat many human diseases, including those caused by altered mitochondrial function or dynamics. Fortunately, the interest in natural products derived from deep-sea species is growing and will continue to do so in the next few years [123].

Despite yielding great advances in the field, several challenges still remain regarding the identification, characterization, and biomedical translation of novel marine organism-derived compounds acting on mitochondria. It is important to highlight that, often, the molecular mechanisms underlying the beneficial effects of many marine derived molecules that can potentially harness mitochondrial function and act as novel therapeutic entities are still unknown. As an example, a huge number of molecules derived from marine organisms have been described for their antioxidant effect; thus, it is speculated that they are likely acting on mitochondria, but a deeper knowledge on mitochondrial involvement is missing. An exception to this is represented by aurilides—drugs that induce apoptosis by interfering with mitochondrial dynamics and cristae organization—whose molecular mechanism of action has been exhaustively elucidated [27,85]. However, the molecular-resolution knowledge available for these drugs has not resulted in more applications at the translational level. This issue is further complicated because the experimental data on molecular actors regulating mitochondrial dynamics, as well as data on the strong connection linking mitochondrial dynamics and function, are constantly increasing. Therefore, the identification of specific targets regulating mitochondrial biology is a field characterized by constant growth and remodeling [122].

Another important limitation for translational research in the field, shared by all the bioactive molecules derived from marine organisms, regardless of their molecular mechanism of action, is that the costs of production for these molecules is very high. Consequently, the characterization of the beneficial biomedical properties of many natural compounds derived from marine organisms remains stuck in the preliminary stage of in vitro testing, failing to reach the requirements for the sustainable implementation of in vivo pre-clinical and clinical trials (Figure 3). However, the technical and methodological tools and knowledge needed to produce bioactive metabolites from marine sources on an industrial scale started to emerge during the early 2000s, when research on natural, biomedical compounds attracted interest among scientific communities, as well as pharmaceutical companies. Strategies centered around producing natural compounds from marine bacteria, fungi, or microalgae are based on implementing culture, harvesting, and extraction procedures on a large scale [124]. This is much more complex for invertebrate sources, where only small amounts of pure compounds are achievable. For these organisms, random sampling directly from the natural environment is not acceptable from an ecological point of view. Moreover, the overall naturally available biomass would be sufficient for industrial demand. Aquaculture of medicine-producing marine invertebrates such as sponges, corals, oysters, or mussels represent a kind of solution to this limitation [125]. Importantly, the development of chemical synthesis also supported the strategies of production on an industrial scale, thus overcoming the issue of isolation from biomass. Synthetic strategies include the possibility of modifying the molecule of interest and developing derivatives with less complexity and more manageable properties. Moreover, the recently outlined genome sequencing approaches allow for an understanding of the biochemical mechanisms regulating the biosynthesis of specific compounds, helping their synthetic cloning. Synthetic biology approaches can also facilitate the generation of genetically modified microbial cell factories to produce heterologous bioactive molecules, including marine organism-derived compounds [126].

The process of bringing a drug candidate to the market requires extensive pre-clinical studies and clinical trials to determine the optimal doses, safety, and efficacy of the drug before it is approved by the FDA or the EMA. High costs in terms of money and time explain why only a handful of promising bioactive molecules will eventually reach the market and consumers. From the data gathered in this review, clearly, the steps connecting the in vitro with the in vivo studies represent the main bottleneck in developing novel therapeutic techniques for mitochondrial illnesses using marine-derived compounds. A strong lack of extensive clinical trials hampers translational investigation in the field. This is the case for many marine-derived compounds that have shown bioactive properties to slow down cancer progression, mainly acting on proliferation. Most studies have been performed in vitro, without taking into account unintended secondary toxic effects on neighboring non-tumor cells or any systemic and non-systemic toxicity. Therefore, we are still far from assessing any safety, dosage, and efficacy issues in human clinical trials and achieving the establishment of these types of compounds for use in precision medicine for patients affected by specific rare diseases.

Beyond in vitro 2D cell cultures, new and powerful tools to rapidly assess the potential efficacy of novel drugs are represented by three-dimensional (3D) organoids, which are emerging as promising models for precision medicine. They can be established rapidly and accurately mimic a patient’s response to therapy [127]. However, the literature describing organoids as a tool to test marine-derived compounds acting on mitochondrial biology is practically non-existent. We are convinced that using human organoids to create a first step of selection for the most promising marine organism-derived compounds to be used to regulate mitochondrial biology in human pathologies before going into organismic models may represent a useful way to accelerate biomedical translation.

This review underscores the importance of focusing on marine organism-derived compounds acting on mitochondrial function to design new treatments against a number of human pathologies with mitochondrial implications. They may act either alone or as adjuvants to gold-standard therapies, exerting synergistic effects in conjunction with conventional drugs. In this field of research, strong efforts should be made to develop well-controlled preclinical trials (in vivo in animal models and/or in vitro in human organoids) as well as clinical trials to assess the huge potential of marine agents in the prevention, treatment, and management of a wide range of human diseases with complex pathologies involving mitochondrial alterations, including neurodegeneration, metabolic diseases, and cancer.

## Figures and Tables

**Figure 1 ijms-25-00834-f001:**
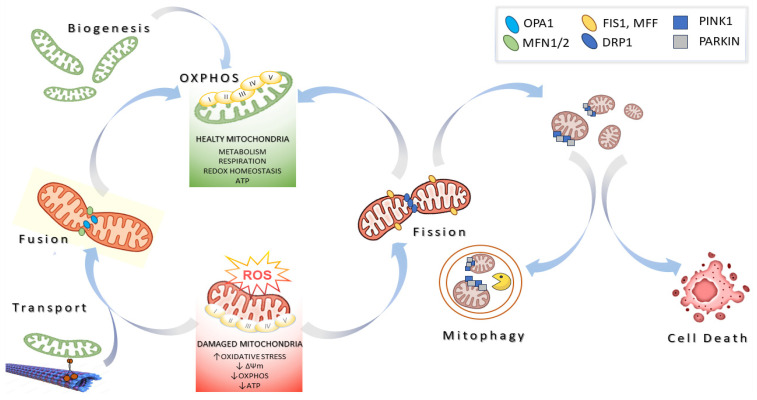
A schematic illustration of mitochondrial dynamics. Mitochondria biogenesis produces a new pool of healthy mitochondria, hubs of cell metabolism and respiration that are central to redox homeostasis and the production of ATP via the OXPHOS complexes I-IV. Mitochondria are highly dynamic; they move within cells along microtubules and undergo the dynamic processes of fusion and fission, as well as turnover via mitophagy. Mitochondria accumulate damage over time. In the case of slight damage, mitochondrial health can be restored by fusion with other mitochondria. On the other hand, in the case of severe damage, dysfunctional mitochondria undergo fission, and the smaller mitochondria are segregated and tagged to be recognized by the macroautophagic mechanism, thus being degraded by mitophagy to avoid cell death. If mitochondrial homeostasis is definitively impaired, cell death pathways are activated. The main molecular players in regulating mitochondrial dynamics are shown (↑ indicates increase and ↓ indicates decrease). MFN1/2: MITOFUSIN1/2; DRP1: dynamin-related protein 1; OPA1: optic atrophy protein 1; FIS1: protein fission 1; MFF: mitochondrial fission factor, PINK: PTEN-induced kinase 1; Δψm: mitochondrial membrane potential.

**Figure 2 ijms-25-00834-f002:**
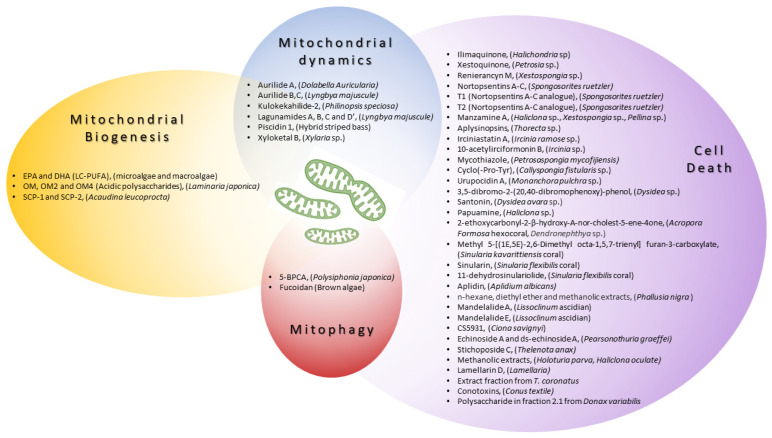
A schematic representation of compounds derived from marine organisms described as biomodulators of specific mitochondrial processes (mitochondrial biogenesis and dynamics) or cellular events involving mitochondria (mitophagy and cell death). Special emphasis should be placed on the modulators of cell death because of their potential as anticancer drugs.

**Figure 3 ijms-25-00834-f003:**
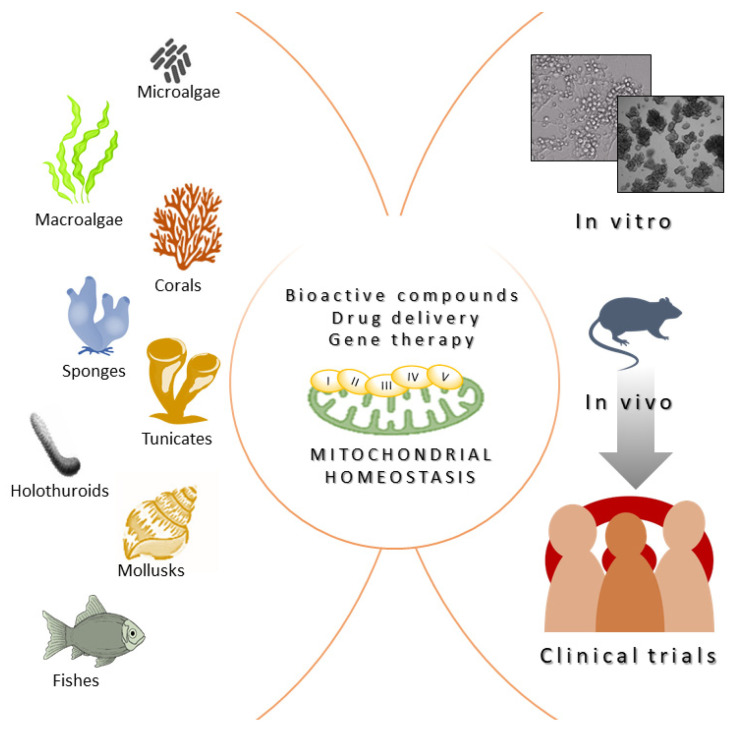
Marine organisms provide beneficial bioactive molecules that can modulate mitochondrial biology and recover mitochondrial homeostasis in pathological contexts. Bioactive compounds, molecules able to facilitate drug delivery, and marine species-derived therapeutic genes need to be tested in vitro (2D and 3D cell and organoid models) and in vivo before being involved in clinical trials with the ultimate aim of ameliorating patients’ quality of life.

**Table 1 ijms-25-00834-t001:** Mitochondrial modulators from marine resources acting on mitochondrial biogenesis, mitochondrial dynamics, and mitophagy. ↑ indicates increase and ↓ indicates decrease.

Compound(s)	Marine Organism	Mechanism of Action Regarding Mitochondria	Cell Line or Model of Disease Used in Preclinical Studies
Mitochondrial Biogenesis			
n-3 PUFA eicosapentaenoic acid (EPA) and docosahexaenoic acid (DHA)	Microalgae and macroalgae	↑PGC1-α, ↑NRF1, ↑mitochondrial biogenesis	C57BL/6J epididymal fat [24]
oligomannuronate (OM) and OM-chromium (III) complexes (OM2 and OM4)	*Laminaria japonica*	↑PGC1-α, ↑mitochondrial function, ↑mitochondrial biogenesis	C2C12, 3T3-L1 [25]
SCP-1 and SCP-2	*Acaudina leucoprocta*	↑AMPK/PGC1-α, ↑NRF2, ↑mitochondrial biogenesis, ↓oxidative stress	Fatigue test in ICR mice [26]
Mitochondrial dynamics			
Aurilide A	*Dolabella auricularia*	Accelerate OPA-1 processing, mitochondrial fragmentation, and the release of CytC [27]	HeLa S3, NCI60 panel [28]
Aurilide B, C	*Lyngbya majuscule*		HCT-8, P388, A549, SK-OV-3, PC-3 [29]
Kulokekahilide-2	*Philinopsis speciosa*		P388, SK-OV-3, MDA-MB-435 [30]
Lagunamides A, B, C and D	*L. majuscule*		P388, A549, PC3, HCT8, SK-OV3, HCT8, MCF7 [31,32]
Piscidin-1	Hybrid striped bass	↓MFN1, ↓MFN2, ↓OPA1, ↓OXPHOS, ↑DRP1, ↑FIS1, ↑mtROS, mitochondrial dysfunction, apoptosis	MG63 [33,34]
Xyloketal B	*Xylaria* sp.	↑Drp1, ↓mitochondrial fregmentation, ↓mitochondrial superoxide production	In vitro model of ischemic stroke in PC12 [35]
Mitophagy			
5-BPCA	*Polysiphonia japonica*	The preservation of PARKIN expression and stabilization of mitochondrial morphology	Model of palmitate (PA)-induced lipotoxicity in a rat pancreatic β-cell line (Ins-1 cells) [36]
Fucoidan: treatment with Fucoidan nanoparticles loaded with proanthocyanidins	Brown algae	↑PINK1, ↑PARKIN, ↓mtDNA release	A model of cisplatin-induced damage in vitro (HK-2 cells) and in vivo (Kunming mice) [37]

**Table 2 ijms-25-00834-t002:** Anticancer effects of compounds derived from marine fauna in different experimental systems. ↑ indicates increase and ↓ indicates decrease.

Compound(s)	Marine Organism	In Vitro/In Vivo Models	Mechanism of Action Regarding Mitochondria	Disease Area
Ilimaquinone	*Halichondria* sp.	MCF-7, MDA-MB-231	caspase activation, ↑ROS, ↓Δψm	Breast cancer [38]
Xestoquinone	*Petrosia* sp.	Molt-4, K562, Sup-T1	↑ROS, ↓HSP90	Leukemia [39]
Renieranycin M	*Xestospongia* sp.	H460	↑BAX, ↓MCL1, ↓BCL2, caspase activation	Lung cancer [40,41]
Nortopsentins A-C	*Spongosorites ruetzler*	P388 cells	caspase activation	Leukemia [42]
T1 (Nortopsentins A-C analogue)	*Spongosorites ruetzleri*	HCT-116 colorectal cancer cells	caspase activation, ↑mitochondrial trans-membrane potential	Colon cancer [43]
T2 (Nortopsentins A-C analogue)	*Spongosorites ruetzleri*	HCT-116 colorectal cancer cells	caspase activation, ↑mitochondrial trans-membrane potential	Colon cancer [42]
Manzamine A	*Haliclona* sp., *Xestospongia* sp., *Pellina* sp.	HCT116 cells	↓BCL2, Δψm loss, ↑caspase activation, CytC release,	Colon cancer [44]
Aplysinopsins	*Thorecta* sp.	K562 cells	↓BCL2, Δψm loss	Leukemia [45]
Irciniastatin A	*Ircinia ramose* sp.	Jurkat cells	↑ROS, ↑JNK, ↑p38, apoptosis	Leukemia [46]
10-acetylirciformonin B	*Ircinia* sp.	HL 60 cells	↓BCL2, ↓Bcl-xL, ↑BAX, ↑ROS, CytC release, apoptosis	Leukemia [46]
Mycothiazole	*Petrosaspongia mycofijiensis*	T47D cells	↓HIF-1 signaling, ↓mitochondrial function	Breast tumor [47]
Cyclo(-Pro-Tyr)	*Callyspongia fistularis* sp.	HepG2 cell	↓BCL2, ↑BAX, ↑ROS, apoptosis	Hepatocellular carcinoma [48]
Urupocidin A	*Monanchora pulchra* sp.	PCa cells	Δψm loss, ↑ROS, CytC release, apoptosis	Prostate cancer [49]
3,5-dibromo-2-(20,40-dibromophenoxy)-phenol	*Dysidea* sp.	PANC-1	Complex II inhibition	Pancreatic carcinoma [50]
Santonin	*Dysidea avara* sp.	ALL B-lymphocytes	↓Δψm, ↑ROS, CytC release, apoptosis	Acute lymphoblastic leukemia [51]
Papuamine	*Haliclona* sp.	MCF-7	mitochondrial damage and JNK activation	Breast cancer [52]
2-ethoxycarbonyl-2-β-hydroxy-A-nor-cholest-5-ene-4one	*Acropora Formosa* hexocoral, *Dendronephthya* sp.	A549	↓ TNF-α, ↓IL-8, ↓Bcl2, ↓MMP2, ↓MMP9, ↑ROS, ↑ BAX, ↑p21, CytC release	Lung cancer [53]
Methyl 5-[(1E,5E)-2,6-Dimethyl octa-1,5,7-trienyl] furan-3-carboxylate	*Sinularia kavarittiensis* coral	THP-1	↓Bcl-xL, ↑BAX, ↑ROS, ↓Δψm, CytC release, apoptosis	Leukemia [54]
Sinularin	*Sinularia flexibilis* coral	SK-HEP-1	↑ROS, ↓Δψm,↓OXPHOS, apoptosis	Liver cancer [55,56]
11-dehydro-sinulariolide	*Sinularia flexibilis* coral	Ca9-22	∆Ψm loss, ↑caspase-3/-9 ↑Bax, ↓Bcl-2/Bcl-Xl, CytC release, apoptosis	Melanoma [55]
Aplidin	*Aplidium albicans*	MOLT-4, NIH3T3	↑ROS, ↓Δψm, ↓ATP, apoptosis	Leukemia, Lymphoma [57,58]
n-hexane, diethyl ether and methanolic extracts	*Phallusia nigra*	Isolated mitochondria from skin tissue of melanoma induced albino/Wistar rats	mitochondrial swelling, ↑ROS, ↓Δψm, CytC release, apoptosis	Melanoma [59]
Mandelalide A	*Lissoclinum* ascidian	NCI-H460, Neuro-2A, HeLa cells	complex V inhibition, apoptosis	Lung cancer, Neuroblastoma [60]
Mandelalide E	*Lissoclinum* ascidian	NCI-H460, HeLa, U87-MG, HCT116	apoptosis	Lung cancer, Glioblastoma [61]
CS5931	*Ciona savignyi*	HCT-8	↑caspase-3, ↑caspase-9, ↑Bax, ↓Δψm, CytC release, apoptosis	Colon cancer [62]
Echinoside A and ds-echinoside A	*Pearsonothuria graeffei*	HepG2, mice	apoptosis	Hepatocarcinoma [63]
Stichoposide C	*Thelenota anax*	HL-60, K562, THP-1, NB4, SNU-C4, HT-29, CT-26; mouse CT-26 subcutaneous tumor and HL-60 leukemia xenograft models	↑Fas, ↑caspase-3, ↑caspase-8, cleavage of Bid, mitochondrial damage, apoptosis	Leukemia, Colorectal cancer [64]
Methanolic extracts	*Holoturia parva*, *Haliclona oculate* sp.	Mitochondria isolated from a rat model of hepatocellular carcinoma	↑ROS, ↓Δψm, CytC release, ↑caspase-3, apoptosis	Hepatocellular carcinoma [65]
Lamellarin D	*Lamellaria*	p388	↓Bcl-2, ↓Δψm, ↑caspase-3, ↑caspase-9, apoptosis	Leukemia [66,67]
Extract fraction of *T. coronatus*	*Turbo coronatus*	EOC cells	↑ROS, ↓Δψm, CytC release, mitochondrial swelling, apoptosis and necrosis	Epithelial ovarian cancer [68]
Conotoxins	*Conus textile*	U87MG	↑ROS, ↓Δψm, CytC release, ↑caspase-3, ↑caspase-9, ↑Bax/Bcl-2	Glioma [69]
Polysaccharide in fraction 2.1	*Donax variabilis*	A549	Mitochondrial disfunction, ↓Δψm, CytC release, ↑caspase-3, ↑caspase-9, ↑Bax/Bcl-2, apoptosis	Lung cancer cells [70]

## Data Availability

Not applicable.

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
