# Peer review of "From Beach to the Bedside: Harnessing Mitochondrial Function in Human Diseases Using New Marine-Derived Strategies"

_ijms, 2024, doi:10.3390/ijms25020834_

Round 1
Reviewer 1 Report
Comments and Suggestions for Authors
Dear authors,
I hope this message finds you well. I have completed the review of your manuscript titled "From Beach to the Bedside: harnessing mitochondrial function in human diseases using new marine-derived strategies." I appreciate the depth of your work and its potential significance in the field.
I am afraid that you still need to make several minor amendments to improve your reviews quality and clarity. Please read the summary and suggestions to improve your manuscript quality listed below.
Summary: The manuscript is well-written and provides a thorough review of biomodulators of mitochondrial physiology from marine sources. Acknowledging your own observations, the manuscript could benefit from addressing the lack of clinical trials and evidence supporting the direct translation of findings into medical practice. But I guess we will all have to wait until more data is generated by world scientists.
Suggestions:
Ensure uniform referencing style throughout the manuscript.
In line 34, consider adding a mention of mitochondria's role in regulating cellular magnesium (Mg2+) levels.
In line 49, enhance clarity by adding "so-called antioxidant" after "cytosolic molecular systems."
In lines 166-168, consider adding information on de novo synthesis vs fission.
In line 186, correct to "largely present in fish" (not fishes).
In line 205, remove one of the occurrences of "ofs."
From line 210 onwards, address the discrepancy in references mentioned and provided. Include the missing reference mentioned at the beginning of the paragraph.
Additional Suggestion: Consider enhancing the clarity of the manuscript by incorporating tables summarizing the role of the identified biomodulators outside the cancer field, specifically addressing chapters 2.2.1 to 2.2.5. This addition will provide readers with a visual aid, facilitating a more accessible understanding of the diverse roles of these biomodulators in various contexts.
Your attention to these suggestions will contribute to the manuscript's overall quality. Thank you for your commitment to advancing this important area of research. For the upcoming holidays, best of wishes.
Comments on the Quality of English LanguageThe quality of English language is fine. Only minor amendments are required.
Author Response
ANSWERS TO REVIEWER 1
Dear authors,
I hope this message finds you well. I have completed the review of your manuscript titled "From Beach to the Bedside: harnessing mitochondrial function in human diseases using new marine-derived strategies." I appreciate the depth of your work and its potential significance in the field.
I am afraid that you still need to make several minor amendments to improve your reviews quality and clarity. Please read the summary and suggestions to improve your manuscript quality listed below.
Summary: The manuscript is well-written and provides a thorough review of biomodulators of mitochondrial physiology from marine sources. Acknowledging your own observations, the manuscript could benefit from addressing the lack of clinical trials and evidence supporting the direct translation of findings into medical practice. But I guess we will all have to wait until more data is generated by world scientists.
We thank the Reviewer for his/her comments. We corrected the errors detected by him/her. Moreover, we added the missing information according to his/her suggestions, enhancing the clarity and completeness of the manuscript.
Suggestions:
Ensure uniform referencing style throughout the manuscript. DONE
In line 34, consider adding a mention of mitochondria's role in regulating cellular magnesium (Mg2+) levels. DONE, see line 35.
In line 49, enhance clarity by adding "so-called antioxidant" after "cytosolic molecular systems." DONE, see line 50.
In lines 166-168, consider adding information on de novo synthesis vs fission. DONE, see lines 169-177.
In line 186, correct to "largely present in fish" (not fishes). DONE, see line 192
In line 205, remove one of the occurrences of "ofs." DONE.
From line 210 onwards, address the discrepancy in references mentioned and provided. Include the missing reference mentioned at the beginning of the paragraph.
We apologize for the mistake related to the referencing style. In this revised version of the manuscript, we uniformed the referencing format, specifically for the following references:
Chen et al., 2023 (line 193)
Wang et al., 2021 (line 250)
Zuo and Kwok, 2021 (line 345)
Fakhri et al., 2022; Wang et al., 2020 (line 365)
Ruiz-Torres et al., 2017 (line 368)
Chen et al., 2012 (line 483)
Additional Suggestion: Consider enhancing the clarity of the manuscript by incorporating tables summarizing the role of the identified biomodulators outside the cancer field, specifically addressing chapters 2.2.1 to 2.2.5. This addition will provide readers with a visual aid, facilitating a more accessible understanding of the diverse roles of these biomodulators in various contexts.
Following the reviewers’ suggestions, we have added a new Table. In this new version, previous Table 1 -summarizing biomodulators from marine organisms aimed to target cancerous cells– is now Table 2. The new Table 1 summarizes marine organism-derived drugs acting on mitochondrial biogenesis, dynamics and mitochondrial elimination by mitophagy. Moreover, to further improve the appeal of the message for the readers and attract their attention, we added a new figure (now Figure 2) with a schematic representation of the bioactive compounds mentioned in the text and modulating specific cellular processes involving mitochondria (mitochondrial biogenesis, dynamics, mitophagy and cell death).
Comments on the Quality of English Language
The quality of English language is fine. Only minor amendments are required.
We carefully reviewed the quality of English Language and introduced minor editing and typo revision to further improve the quality of the manuscript.

Reviewer 2 Report
Comments and Suggestions for Authors
This long and interesting review makes a sincere effort to link two seemingly distant subfields of translational medicine, each heavy on its own: seeking therapeutics for mitochondrial malfunctioning in marine animals-derived pharmaceutical products. Selecting this topic immediately implies the need to dive into the two topics separately, which the authors do; however, it also calls for finding new creative ways to highlight the presented links and attract the readers’ attention to the entire text, which calls for improvements. One option in that direction can be by adding more figures, each focused on another subsection or another subtype of proposed drugs aimed to target a particular part of the mitochondrial process. This can be a substantial added value.
Language and editing wise, the text may benefit from improvement as well. Specifically, some pages appear to include a single paragraph, which complicates the reading; and a polishing act by a native English user can be of assistance as well. The material is of high quality and extensive value, it deserves the effort to make it more reader-friendly as well.
Comments on the Quality of English LanguageModerate editing of English language required.
Author Response
ANSWERS TO REVIEWER 2
This long and interesting review makes a sincere effort to link two seemingly distant subfields of translational medicine, each heavy on its own: seeking therapeutics for mitochondrial malfunctioning in marine animals-derived pharmaceutical products. Selecting this topic immediately implies the need to dive into the two topics separately, which the authors do; however, it also calls for finding new creative ways to highlight the presented links and attract the readers’ attention to the entire text, which calls for improvements. One option in that direction can be by adding more figures, each focused on another subsection or another subtype of proposed drugs aimed to target a particular part of the mitochondrial process. This can be a substantial added value.
We thank the Reviewer for his/her comment. We totally agree about the importance of grabbing the readers ‘attention to the entire text, and therefore have added a new figure and table.
Following the reviewer's suggestion, a new figure (now Figure 2) with a schematic representation of the compounds mentioned in the text and described as biomodulator of specific mitochondrial processes (mitochondrial biogenesis and dynamics) or cellular events involving mitochondria (mitophagy and cell death) has been incorporated to the article.
Moreover, we added a second table (now Table 1) to the preexisting table (now Table 2) to summarize respectively: 1) biomodulators acting on mitochondrial biogenesis, dynamics and mitochondrial elimination by mitophagy, and 2) biomodulators identified as modulator of cell death and thus applied in the cancer field.
Language and editing wise, the text may benefit from improvement as well. Specifically, some pages appear to include a single paragraph, which complicates the reading; and a polishing act by a native English user can be of assistance as well. The material is of high quality and extensive value, it deserves the effort to make it more reader-friendly as well.
Comments on the Quality of English Language
Moderate editing of English language required.
We carefully reviewed the quality of English Language and introduced minor editing and typo revision to further improve the quality of the manuscript.

Round 2
Reviewer 2 Report
Comments and Suggestions for Authors
I find the revised manuscript acceptable in its present form